

# COVID-19 and diet: efforts towards a balanced diet and sustainable nutrition among university students in Pakistan

Zeshan Ali[1], Lienda Bashier Eltayeb[2], Sndos Z.A. Fattiny[3], Iftikhar Younis Mallhi[4], Farah Javed[4], Ali Abdullah Alyousef[5], Qandeel Ijaz[6], Shoaib Younas[7], Ishrat Khan[8], Zahra Batool[9] and Muhammad Ahmad[10]

[1] College of Food Science and Engineering, Bohai University, Jinzhou, China
[2] Department of Medical Laboratory Science, Prince Sattam Ibn Abdelaziz University, Riyadh, Saudi Arabia
[3] Department of Food Science and Nutrition, College of Food and Agriculture Sciences, King Saud University, Riyadh, Saudi Arabia
[4] Department of Food Science and Human Nutrition, University of Veterinary and Animal Sciences, Lahore, Pakistan
[5] Department of Clinical Nutrition, Mental Hospital Al-Ahsa, Minister of Health, Hofuf, Saudi Arabia
[6] Department of Social Sciences, University of Punjab, Lahore, Pakistan
[7] Department of Food Technology, University of Central Punjab, Lahore, Pakistan
[8] Department of Biotechnology, Federal Urdu University of Arts, Science and Technology, Karachi, Pakistan
[9] Department of Biosciences, COMSATS University, Sahiwal, Pakistan
[10] Department of SST, Information Security, University of Management & Technology, Lahore, Pakistan

Corresponding author
Lienda Bashier Eltayeb,
l.eltayeb@psau.edu.sa

## ABSTRACT

Nutrition is an important aspect of a healthy lifestyle for all individuals, including adolescents. The objective of this cross-sectional descriptive survey study was to investigate university students' awareness of immunity enhancing foods, food nutritive values, and eating practices during the COVID-19 pandemic. A total of 839 university students from four different universities in Pakistan participated in the study from October 2021 to January 2022, 397 of which were male and 442 were female. A total of 419 students were studying in medical disciplines while 420 were non-medical students. The students had significant knowledge ($p < 0.05$) about COVID-19, and nutritional habits were seen in both medical and non-medical students. Results showed that medical students (n-201) were slightly more aware of immunity-enhancing foods and the nutritive values of foods compared to engineering students (n-79). However, eating practices were generally poorly adopted by all of the university students. Male and female students were not significantly different in their mean replies to questions on the nutritive value of food or in their eating habits. Healthy eating practices will aid university students in preventing illnesses connected to nutrition as well as enhancing their immune systems and nutritional well-being both during and post-pandemic. In light of these results, suggestions and implications for nutritional advice and education were explored.

## INTRODUCTION

The coronavirus disease 2019 (COVID-19) pandemic spread across the world only a few months after its initial cases were detected in Wuhan, China (*Alagawany et al., 2021*; *Ali et al., 2021*). In Pakistan, the first case was reported on 26 February 2020, and it continued to spread through the whole country by 11 March 2020. According to the latest reports by the World Health Organization (WHO) (*WHO, 2023*), there have been 1,580,631 confirmed cases of COVID-19 with 30,656 fatalities reported between 3 January 2020 and 4 October 2023 in Pakistan. A total of 340,016,117 doses of vaccine have been given as of 19 September 2023. The WHO declared the COVID-19 outbreak as a global pandemic because of its spread in several countries along with China (*Caso et al., 2022*). The causative agent for COVID-19 is coronavirus which belongs to the enveloped RNA virus family and is also notorious for causing outbreaks of severe SARS-CoV in 2002 and MERS-CoV (*Alhusseini et al., 2021*; *Anis et al., 2021*).

As for the mode of transmission, MERS-CoV was reported to spread from camels to humans while SARS-CoV and COVID-19 seemed to originate from bats, starting with the first report from Wuhan, China, and later spread all over the globe through human-human transmission (*Modi et al., 2020*). The symptoms of COVID-19 infection appear after an incubation period of about 5 days and include high temperature, throat problems (*e.g.*, dry cough), weakness, breathing problems, and sometimes gastrointestinal symptoms (*Pendyala & Patras, 2020*). Clinical signs and symptoms range from mild to severe (*e.g.*, respiratory distress syndrome and various organ failures). Infection can progress to pneumonia, respiratory failure, and death if left untreated (*Clemente-Suárez et al., 2021*).

Although health workers and researchers across the globe are trying to fight the pandemic through vaccines as well as various public health initiatives, a new variant emerges every few months and poses a global threat, but predominantly to people with comorbidities and weak immune systems, which is highly influenced by the nutritional status of the individual at risk. A multitude of studies have found that different factors like age, lifestyle, physical activities, nutritional status, and dietary habits greatly affect the immune system of an individual (*Aman & Masood, 2020*; *Naja & Hamadeh, 2020*). Other studies have shown favorable changes in individuals' eating habits in response to lockdown, such as an increase in the number of meals prepared and consumed at home (*Guicciardi & Pazzona, 2020*). However, other studies showed contradictory results including an increase in alcohol consumption, enhanced consumption of comfort food due to emotional distress (*Guicciardi & Pazzona, 2020*), and reduced sleep quality (*Bakhsh et al., 2021*). The reasons for these changes include having a more sedentary lifestyle due to increased screen time (*Ramalho, 2020*), less time working, less walking to public transportation or to work (*Salazar-Fernández et al., 2021*), and not being engaged in physical activities (*Lima et al., 2021*). Dietary pattern has a strong association with anxiety and depression since multiple factors, namely ailment, socioeconomic status, behaviors, emotions, and cultural pressures, can affect the dietary intake of an individual. Additionally, these modifications have an impact on a person's physical activity (*Cena & Chieppa, 2020*). Social connectedness, dietary patterns, anxiety,

and depression are linked to one another and are significantly influenced by changes in the environment. Social isolation impacts a person's health by causing changes in eating times, sleeping pattern, and physical activity, which can majorly increase sedentary behavior that leads to an increased risk of obesity. Fear and anxiety, common outcomes of quarantine, may also cause changes in dietary habits leading to unhealthy dietary patterns (*Aman & Masood, 2020*; *Mentella et al., 2021*).

Nutrition and immunity have a robust connection and only a balanced diet will ensure a strong immune system that can fight against viruses, and specific foods or combination of foods can have an effect on gene expression. Nutritional deficits lead to impaired immune response and increased vulnerability to infection (*Aman & Masood, 2020*; *Mentella et al., 2021*). This study was conducted in order to assess the knowledge of university students on COVID-19 and its associations with diet. Ideas and consequences for nutritional counseling and education were then examined in light of these findings.

## MATERIALS AND METHODS

### Study design and setting

The study adopted a cross-sectional descriptive survey design and was carried out in four different universities from two developed and two non-developed cities of Pakistan. The study was conducted in accordance with the Declaration of Helsinki and was approved by the Institutional Review Board of Barani Institute of Sciences Sahiwal, Pakistan (IRB# 2001).

### Participants

Study participants were selected from a population of 10,000 students across four different universities (main campuses and sub campuses) in Pakistan from October 2021 to January 2022. Initially, the study participants were 1,000 selected medical and engineering students. Out of the 1,000 participants, 839 students completed the study. The inclusion criteria for the study subjects included written consent, availability to complete and return the questionnaire to the researcher on the spot, and status as a regular university student. People who were not currently enrolled as university students, had a history of serious eating disorders, or who refused to answer the survey truthfully and consistently were all excluded.

### Assessment tools

The study questionnaire consisted of basic and widely-used tools validated in English in order to assess the important pillars of lifestyle medicine: medical history, physical activity, COVID-19 information, food and nutrition history, eating pattern (including risky substances such as smoking), and future nutrition goals. In addition, it asked about the participants' self-reported height, weight, and smoking status as well as their sociodemographic factors.

### Data collection

Data were collected over four months (October 2021 to January 2022) using a direct delivery and retrieval strategy implemented by the researchers. Researchers met the enrolled students

individually and gave them the questionnaire to fill out and return directly. Incomplete forms were removed from the study. In order to reach as many students from the different universities as possible, instructors from all colleges were asked for permission to invite students from their classes to take the survey titled "Nutrition and Health Behaviour." The last part of the survey consisted of an approved consent form; in other words, consent was obtained. Students were able to change their minds and withdraw from the study at any time, for any reason, without imposing any penalties or losing their eligibility to receive any future services from the institution.

The survey consisted of seven groups of questions around health and wellness, and requested information including the participants' gender, age, ethnicity/race, year at college, medical history, COVID-19 information, eating habits, frequency of eating, types of daily beverages, appetite behavior, weight history, satisfaction about appearance, and nutritional goals.

### Power estimation
To estimate the sample size, the power analysis method was conducted using SPSS software version 20. The results showed that the power of the current survey study was 0.90.

### Data analysis
Descriptive analysis (mean and SD) was used to analyze the data. Item scores were designated as dependent variables and the medical and non-medical factors were set as independent variables. Using purposively set benchmark mean values, the item scores of the first cluster of questionnaire section A (medical history of participants) were interpreted on a scale of Yes or No. The item scores of the second cluster of questionnaire section B (information regarding COVID-19) were interpreted as Not Aware (NA), Slightly Aware (SA), Moderately Aware (MA), and Extremely Aware (EA). The item scores for the third cluster of questionnaire section C (food and nutrition information) were interpreted as Fully Aware (FA), Moderately Aware (MA), and Not Aware (NA). The item scores for the fourth cluster of questionnaire section D (eating practices of university students) were interpreted as Fully Adopted (FA), Moderately Adopted (MA), and Not Adopted (NAd); 2 T = two times, DS = Drink + Soda. The scores of the fifth cluster of questionnaire section E (weight history) were interpreted as Scale: Yes or No, Gained Weight (GW), Weight Loss (WL), Satisfied (S), Very Satisfied (VS), Dissatisfied (DS), and Very Dissatisfied (VDS). The scores for the sixth cluster of questionnaire section F (physical activity history) were interpreted as Physically activity Scale (Yes or No), Rate of workout intensity Light (L), Moderate (M), and Vigorous (V). The scores for the seventh cluster of questionnaire section G (nutritional goals) were interpreted as Important or Unimportant and in this section, participants were also asked about data publication on a Yes or No scale. The Student t test was employed to investigate the differences between the medical and nonmedical students. The Shapiro–Wilks normality test was used to determine the distribution of the data's normality before the t tests were run. There was a normal distribution of the data ($p$ −0.70). Item scores were utilized as test variables for the t tests, with medical and non-medical being the grouping variable. When coding, the label for medical students was

**Table 1  Demographic characteristics of study participants.**

| | Characteristics | Medical students (*n*) | Non-medical students (*n*) |
|---|---|---|---|
| Gender | Male | 168 | 229 |
| | Female | 251 | 191 |
| University Name | GCUF | 100 | 100 |
| | Barani | 100 | 120 |
| | Montgomery | 109 | 100 |
| | UCP | 110 | 100 |
| Age (Years) | 18–25 | 347 | 376 |
| | 26–35 | 59 | 36 |
| | 36–45 | 13 | 8 |
| Education level | Bachelor's degree | 378 | 397 |
| | Master's degree | 41 | 23 |
| | Total | 419 | 420 |

**Notes.**
UCP, University of Central Punjab; GCUF, Government College University Faisalabad.

assigned the numerical value 1, while the label for non-medical students was assigned the numerical value 2.

# RESULTS

## Participant characteristics

A total of 839 participants completed questionnaires and were included in the final analysis. The remaining 161 forms were excluded from the study because of incomplete data. Table 1 shows the demographic characteristics of all participants (gender, age, university name, and education level). Of the total number of participants (839), 49.9% were medical and 50.1% were non-medical students, and came from the following universities: GCUF Medical (11.9%), Barani Medical (11.9%), Montgomery Medical (13.0%), UCP Medical (13.1%), GCUF Engineering (11.9%), Barani Engineering (14.3%), Montgomery Engineering (11.9%), and UCP Engineering (11.9%).

## Medical history of participants

Table 2 shows the average value of questionnaire results asked about the medical history of the participants. The average answers from medical and non-medical students of all universities about having any food allergy or intolerance, taking any type of supplement, and smoking were No. The average answer of all participants about having any past or current medical conditions was Yes. Overall, 55.6% of medical students were shown to have a history of past illness, while 47.85% of non-medical students showed a past history of illness. Medical students showed more cases of food intolerance as compared to non-medical students, *i.e.,* 39.14% and 15.95%, respectively. Non-medical students showed a higher percentage of smoking compared to medical students, *i.e.,* 52.61% and 45.82%, respectively.

**Table 2 Information about medical history of participants (Scale: Yes).**

| Medical history | Medical students (*n*-419) | | Non-medical students (*n*-420) | |
|---|---|---|---|---|
| | *n* | % | *n* | % |
| Do you currently or previously have any medical illnesses for which you are receiving treatment? | 233 | 55.60 | 201 | 47.85 |
| Do you have any food intolerances or allergies that have been medically diagnosed? | 164 | 39.14 | 67 | 15.95 |
| Do you use any dietary, herbal, mineral, or sports supplements? | 205 | 48.92 | 196 | 46.66 |
| Do you smoke? | 192 | 45.82 | 221 | 52.61 |
| Do you feel stressful in daily life? | 188 | 44.86 | 201 | 47.85 |

## COVID-19 information among participants

In addition to the medical history of participants, we analyzed data about information regarding COVID-19. Results in Table 3 show the participants' awareness levels of COVID-19 information. Items about COVID-19 were answered on a three-point scale (Fully Aware, Moderately Aware, and Not Aware). On average, compared to non-medical students, medical students were more aware of immunity, immunity playing a vital role in COVID-19 infection, the first initial symptoms of COVID-19, that eating or having contact with wild animals would result in infection by the COVID-19 virus, that general medical masks prevent infection of the COVID-19 virus, and the use of hand sanitizer to wash hands after touching foreign surfaces outside the house and shaking hands.

## Food and nutrition information among participants

Table 4 shows that students, after answering on a three-point scale, were fully aware that vitamin C helps to boost immunity, water helps to maintain homeostasis in the body, and carbohydrates elevate blood sugar levels. Additionally, students had a fair understanding of the importance of protein for the maintenance and repair of body tissues, the energy-giving properties of carbohydrates, the stimulation of insulin production by glucose, the protective properties of foods high in vitamins and minerals, the role of calcium in the maintenance and development of strong bones and teeth, and the necessity of milk and other dairy products for bone health. Although all students were aware of food and nutrient information, medical students showed more understanding when compared to non-medical students (Table 4).

## Eating practices of participants

The results from participants' responses to survey questions about eating habits, frequency of eating, and types of daily beverages are shown in Table 5. On average, eating habits like eating breakfast at home before leaving for work, skipping breakfast and only eating lunch and dinner, eating breakfast at work, preparing breakfast at home and bringing it to work, purchasing lunch from a university cafe, purchasing lunch from restaurants, eating at snack time, consuming soft drinks daily, having dinner at six o'clock each day, and purchasing breakfast or lunch from a restaurant are all unhealthy habits of university students from the

**Table 3** Information regarding COVID-19 among medical students.

| | Information regarding COVID-19 | Not aware | Moderately aware | Fully aware | P-value |
|---|---|---|---|---|---|
| Medical students | Do you know about immunity? | 54 (12.8) | 164 (39.1) | 201 (47.9) | |
| | Do you think immunity plays a vital role in COVID-19 infection? | 96 (22.9) | 138 (32.9) | 185 (44.1) | |
| | First initial symptom of COVID-19? | 110 (26.2) | 102 (24.3) | 207 (49.4) | |
| | Infection by the COVID-19 virus could occur from eating or coming into touch with wild animals? | 79 (18.8) | 131 (31.2) | 209 (49.8) | |
| | COVID-19 virus is spread by infected people's respiratory droplets? | 43 (10.2) | 137 (10.2) | 239 (57.0) | |
| | Do you use medical masks to protect yourself from COVID-19 infection? | 146 (34.8) | 91 (34.8) | 182 (43.4) | 0.001[*] |
| | After touching foreign surfaces outside your home, do you use hand sanitizer? | 91 (21.7) | 66 (21.7) | 262 (62.5) | |
| | Do you still shake hands with people after COVID-19 pandemic? | 21 (5.0) | 19 (4.5) | 379 (90.4) | |
| Non-medical students | Do you know about immunity? | 146 (34.7) | 195 (46.4) | 79 (18.8) | |
| | Do you think immunity plays a vital role in COVID-19 infection? | 148 (35.2) | 174 (41.4) | 98 (23.3) | |
| | First initial symptom of COVID-19? | 45 (10.7) | 61 (14.5) | 314 (74.7) | |
| | Infection by the COVID-19 virus could occur from eating or coming into touch with wild animals? | 76 (18.0) | 143 (34.0) | 201 (47.8) | |
| | COVID-19 virus is spread by infected people's respiratory droplets? | 14 (3.3) | 27 (6.4) | 379 (90.2) | |
| | Do you use medical masks to protect yourself from COVID-19 infection? | 76 (18.0) | 147 (35.0) | 197 (46.9) | 0.003[*] |
| | After touching foreign surfaces outside your home, do you use hand sanitizer? | 81 (19.2) | 154 (36.6) | 185 (44.0) | |
| | Do you still shake hands with people after COVID-19 pandemic? | 72 (17.1) | 147 (35.0) | 219 (52.1) | |

**Notes.**
*Significant values.

cities Sahiwal and Lahore. On average, students typically eat two times a day and consume beverages like soda on a daily basis. Both types of students (medical and non-medical) did not show good eating practices or behavior.

## Weight history of participants

Table 6 shows the results of the participants' appetite behaviors, weight history, and satisfaction about appearance. Results show that the majority of the participants said Yes to questions about recent changes in appetite and concern about weight loss, but No to questions about trying to lose or gain weight. In addition, most participants wanted to lose weight and were dissatisfied with their appearance. Appetite changes and weight gain or loss practices were seen more in non-medical students. Medical students were more worried about their weight compared to non-medical students. Overall, medical students were not satisfied with their current body looks or appearance. Table 6 shows the physical

**Table 4  Food and nutrition information.**

| | Knowledge about food & nutrition | Not aware | Moderately aware | Fully aware | P-value |
|---|---|---|---|---|---|
| **Medical students** | Body tissues require protein to be built and repaired | 103 (24.5) | 107 (25.5) | 209 (49.8) | |
| | The body gets its energy from carbs | 76 (18.1) | 117 (27.9) | 226 (53.9) | |
| | Blood sugar levels are elevated by carbs | 44 (10.5) | 137 (32.6) | 238 (56.8) | |
| | Insulin production is stimulated by glucose | 39 (9.3) | 105 (25.0) | 275 (65.6) | |
| | Do you know about omega3 fatty acids? | 155 (36.9) | 164 (39.1) | 100 (23.8) | |
| | Healthy foods are sources of vitamins and minerals | 7 (1.6) | 103 (25.2) | 309 (73.7) | |
| | Vitamin C helps to enhance immunity | 10 (2.3) | 91 (21.7) | 318 (75.8) | |
| | Maintaining calcium homeostasis is aided by vitamin D | 31 (7.3) | 76 (18.1) | 312 (74.4) | 0.004* |
| | Calcium supports healthy bone and tooth development and maintenance | 54 (12.8) | 60 (14.3) | 305 (72.7) | |
| | Dairy products, such as milk, are crucial for bone health and water helps the body eliminate waste materials. | 2 (0.4) | 36 (8.5) | 381 (90.9) | |
| | The body's homeostasis is supported by water | 8 (1.9) | 40 (9.5) | 371 (88.5) | |
| | Nutrients are transported by the water to the body cells | 1 (0.2) | 40 (9.5) | 379 (90.4) | |
| **Non-medical students** | Body tissues require protein to be built and repaired | 92 (21.9) | 114 (27.1) | 214 (50.9) | |
| | The body gets its energy from carbs | 97 (23.0) | 126 (30.0) | 197 (46.9) | |
| | Blood sugar levels are elevate by carbs | 76 (18.0) | 143 (34.7) | 201 (47.8) | |
| | Insulin production is stimulated by glucose | 91 (21.6) | 147 (35.0) | 182 (43.3) | |
| | Do you know about omega3 fatty acids? | 314 (74.7) | 77 (18.3) | 29 (6.9) | |
| | Healthy foods are sources of vitamins and minerals | 297 (70.7) | 47 (11.1) | 76 (18.0) | |
| | Vitamin C helps to enhance immunity | 41 (9.7) | 145 (34.5) | 234 (55.7) | |
| | Maintaining calcium homeostasis is aided by vitamin D | 274 (65.2) | 67 (15.9) | 79 (18.8) | 0.004* |
| | Calcium supports healthy bone and tooth development and maintenance | 36 (8.5) | 87 (20.7) | 297 (70.7) | |
| | Dairy products, such as milk, are crucial for bone health and water helps the body eliminate waste materials. | 27 (6.4) | 115 (27.3) | 278 (66.1) | |
| | The body's homeostasis is supported by water | 297 (70.7) | 98 (23.3) | 25 (5.95) | |
| | Nutrients are transported by the water to the body cells | 214 (50.9) | 150 (35.7) | 56 (13.3) | |

Notes.
*Significant values.

activity history reported by the participants. The majority of students were physically active and the average intensity of the students' workout was light.

## Nutritional goals of participants

Table 7 shows the participant's nutritional goals. Participants answered that it was important to make changes in their nutritional habits and it is also important that they are confident in their ability to improve their habits. In addition, participants allowed this research to be published. Medical students showed a positive response in changing eating behavior compared to non-medical students. After comparing the results from the medical and non-medical students, there was no significant difference between medical and non-medical students as the p value for all t tests ranged from 0.47–0.50.

**Table 5  Eating practices of university students (Scale: Yes).**

| Eating practices of students | Medical students | | Non-medical students | |
|---|---|---|---|---|
| | *n* | % | *n* | % |
| Do you skip breakfast and having only lunch and dinner? | 218 | 52.02 | 197 | 46.90 |
| Do you skip lunch? | 113 | 26.96 | 124 | 30.23 |
| Do you eat breakfast at home before going to work? | 87 | 20.76 | 101 | 24.76 |
| Do you eat breakfast at work? | 104 | 24.82 | 96 | 22.85 |
| Do you carry prepared breakfast from home to work? | 76 | 18.13 | 41 | 9.76 |
| Do you buy food at work from university cafe? | 19 | 4.53 | 33 | 7.85 |
| Do you buy food from fast foods restaurants? | 241 | 57.51 | 257 | 61.19 |
| Do you eat at snack time? | 197 | 47.01 | 204 | 48.57 |
| Do you eat balanced meals three times daily? | 172 | 41.05 | 197 | 46.90 |
| Do you buy snacks as lunch? | 34 | 8.11 | 43 | 10.23 |
| Do you consume soft drinks daily? | 201 | 47.97 | 196 | 46.66 |
| Do you have dinner at 6 PM daily? | 12 | 2.86 | 21 | 5.0 |
| Do you buy breakfast or lunch from food vendors? | 07 | 1.67 | 17 | 4.04 |
| Do you consume caffeinated beverages on a regular basis? | 376 | 89.73 | 351 | 83.57 |

**Table 6  Weight history. (Scale: Yes).**

| Questions | Medical students | | Non-medical students | |
|---|---|---|---|---|
| | *n* | % | *n* | % |
| Has your appetite changed recently | 147 | 35.08 | 189 | 45.0 |
| Have you recently gained or lost weight? | 114 | 27.20 | 131 | 31.19 |
| Did you ever worry about your weight? | 274 | 65.39 | 201 | 47.85 |
| Have you previously attempted to gain or lose weight? | 245 | 58.47 | 276 | 65.71 |
| How satisfied are you overall with the way your body looks? | 179 | 42.72 | 223 | 53.09 |
| Do you exercise regularly now? | 291 | 69.45 | 301 | 71.66 |

**Table 7  Nutritional goals (Scale: Yes).**

| Questions | Medical students | | Non-medical students | |
|---|---|---|---|---|
| | *n* | % | *n* | % |
| How essential do you think changing your eating habits is? | 289 | 68.97 | 298 | 71.12 |
| How sure are you that you can change the way you eat? | 114 | 27.20 | 97 | 23.15 |
| Can we publish your data for research purpose? | 419 | 100.0 | 420 | 100.0 |

## DISCUSSION

Our findings demonstrate the impact of COVID-19 on patients' lifestyles and preferences. During lockdown, people consumed unhealthy foods, creating an obesogenic environment. Increased involvement in the kitchen means more people have time to improve their eating habits (*Caso et al., 2022*). Most people have difficulty eating healthy because they lack willpower, have time constraints, and have taste preferences (*Pinho et al., 2018*). It has

been found that lockdowns have a significant impact on people's lives that can result in improvements in nutritional behavior. This study compared medical and non-medical students based on their medical history, knowledge about COVID-19, lifestyle habits, and weight history.

Autoimmune disorders are associated with significant delays in recovery because they damage the immune system, which ultimately causes COVID-19 patients to heal more slowly (*Alagawany et al., 2021*; *Liu & Liu, 2020*). Overall, our results showed that there were no food allergies, supplements, or smoking among the participants. According to our results, smoking had no significant effect on the recovery period. This is in agreement with the study by *Leung, Yang & Sin (2020)* but in contrast with the study by *Reddy et al. (2021)*. The effect of smoking on COVID-19 infection has been found to be controversial when comparing a number of studies (*Cai et al., 2020*; *Leung, Yang & Sin, 2020*; *Reddy et al., 2021*).

The results of this study indicated that students were aware of COVID-19 symptoms and complications. Human-to-human transmission occurs *via* droplets, feco-oral contact, and direct contact, and it takes 2 to 14 days for the disease to develop. In most cases, participants were aware that the infection affects the respiratory system, causing dry coughing and difficulty breathing at high temperatures. For students who are keenly following the COVID-19 situation in their country, this may help them prepare physically and mentally by acquiring the essentials necessary for fighting the disease, as well as social distancing, practicing hand hygiene, avoiding traveling, and using face masks. A number of participants demonstrated their commitment to precautionary measures for virus protection, their awareness of the virus' dangers, and their fear of infection (*Elgendy & Abdelrahim, 2021*).

Based on their food and nutritional history, students were aware of the importance of water, carbohydrate consumption, protein consumption, vitamin D, and omega-3 fatty acids during the COVID-19 recovery process. *de Faria Coelho-Ravagnani et al (2021)* also found benefits by increasing water consumption by 2 liters. Proper nutrition and water play a crucial role in the body's immune defense against COVID-19 (*ASPEN, 2020*). According to a study, nuts are significantly associated with recovery because they are rich in bioactive peptides, immune-supporting antioxidant minerals like zinc and selenium, and anti-inflammatory omega-3 polyunsaturated fatty acids (*Zabetakis et al., 2020*). Fruits with bioactive ingredients, *e.g.*, vitamin C, anthocyanins, and minerals, were also beneficial during recovery (*Pendyala & Patras, 2020*). Due to the riboflavin content in milk and rich microbiotics in yogurt that enhance immunity, milk also appears to provide a level of protection against COVID-19 (Fig. 1).

Both types of students (medical and non-medical) did not show good eating practices or behaviors, which should be a great concern for the community. According to the research, skipping breakfast frequently may weaken a person's immune system. Immunity cells (monocytes) are produced in the bone marrow and are typically seen scouring the body for infections. Additionally, these cells are involved in tissue healing and inflammation. Researchers discovered that immune cells returned to the bone marrow during fasting periods from the bloodstream. Monocytes, however, rushed back into the

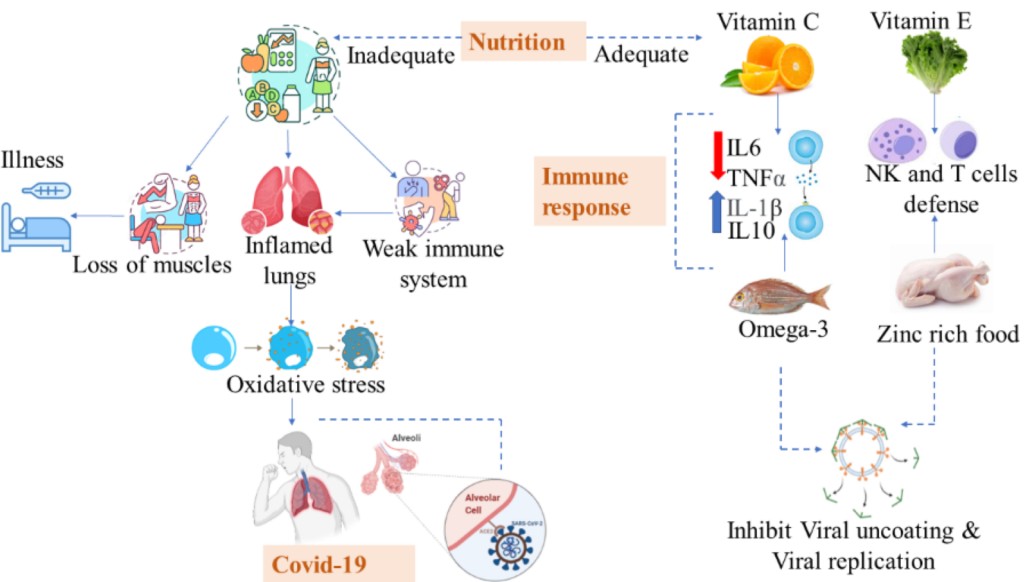

**Figure 1 Adequate and inadequate nutrition in COVID-19.**

bloodstream as soon as feeding was resumed, resulting in monocytosis, an abnormally high quantity of these immune cells. Since circulating monocytes are significant contributors to inflammation, fasting lowers their number, which some would view as positive. However, the reintroduction of food causes a spike in the number of monocytes that return to the blood, which can be dangerous and affect lipid profile parameters (*Ashraf & Ali, 2018*; *Janssen et al., 2023*). We also been found that students consume beverages and soda on a daily basis. Previous studies have confirmed that avoiding unhealthy dietary habits such as eating fast/fried/junk/spicy foods and drinking cold drinks helped people recover from illnesses faster (*Cena & Chieppa, 2020*; *de Faria Coelho-Ravagnani et al., 2021*). Vitamin D deficiency was found in COVID-19 patients, and some studies supported the intake of vitamins and minerals, especially vitamin D. A dietitian or physician should be consulted before taking supplements in order to prevent adverse interactions between food and drugs, drugs and medical treatments, or medical treatments and drugs. As a result of these dietary guidelines, the general population has been focused on improving their diets, as guided by the WHO and FAO (*FAO, 2020*). COVID-19 outcomes can be influenced by nutritional status and diet because they modulate inflammation and immune responses (*Silverio et al., 2021*).

According to this study, weight gain after COVID-19 was not as high as during COVID-19. This is supported by several studies (*Chen et al., 2020*; *Chtourou et al., 2020*). Participants' physical activity habits were analyzed. The majority of participants in this study were physically active. In previous studies, it was suggested that people's physical activity reduced during lockdowns (*Castañeda-Babarro et al., 2020*; *McDowell et al., 2020*). In accordance with *Nizami & Mujeebuddin (2020)*, sleep hours were among the lifestyle activities that significantly affected COVID-19 recovery. An active lifestyle has been

associated with a quicker recovery from COVID-19 infection. Maintaining a healthy immune system includes staying physically active, eating well, managing weight, and managing stress (*Khoramipour et al., 2021*). Similar results were found in a study of COVID-19-infected healthcare professionals who followed a healthy diet and exercise regimen (*Do et al., 2020*).

In this study, participants were aware that making changes to nutrition habits could help in combating COVID-19. In previous studies, it was shown that changes or modifications to dietary habits improved outcomes in COVID-19 patients (*Cena & Chieppa, 2020*; *de Faria Coelho-Ravagnani et al., 2021*; *Rabail et al., 2021*).

### National food safety emergency response plans

Pakistan faces food safety and security challenges, including foodborne illness, contamination, adulteration, and scarcity of food. The Pakistan Food Authority (PFA), established in 2017, plays a central role in regulating and enforcing food safety standards. The National Food Safety Contingency Plans are essential to protect public health and ensure food safety and security in Pakistan. These plans address a range of issues, including foodborne disease outbreaks and contamination incidents, in a coordinated effort between the PFA, the Ministry of Health, and provincial food authorities. They outline procedures for rapid detection, reporting, and response to food safety problems, and include surveillance, laboratory testing, recalls, public awareness campaigns, and communication strategies. Given Pakistan's diverse food landscape, these plans are critical to maintaining a safe and trustworthy food market.

### Study limitations

Our study has a number of limitations that deserve consideration. We did not stratify the population based on BMI, and the questionnaire we utilized was not intended to take energy consumption into account. The physical state of our subjects may have had an impact on food, taste, and eating habits. Finally, since each item was self-reported, reporting bias may have been introduced. Despite these drawbacks, we think that our findings are significant and highlight key changes in a particularly unique population. Our results may assist other researchers in studying the effects of lockdowns and aid in future pandemic preparedness.

### Future research directions

The incorporation of wearable technology and mobile applications for nutritional tracking could improve data accuracy and track participants' eating patterns in real time. Additionally, longitudinal studies that track dietary patterns and their consequences over time can provide important new information about the long-term effects of nutrition on health. A comprehensive understanding of food behavior and its psychological foundations can be fostered by cross-disciplinary studies that integrate dietitians, psychologists, and data scientists.

## CONCLUSION

This article was written to help provide an understanding of the unique challenges facing university students caused by the COVID-19 pandemic, as well as provide strategies to

mitigate them. In order to avoid the spread of misinformation regarding nutrition and dietary intake, as well as the COVID-19 pandemic, individuals should adopt a healthy lifestyle, modify their diet with fruits and vegetables, exercise whenever they have free time, maintain their weight, and get a minimum of 8 h of sleep. Supporting the development of holistic lifestyle solutions is crucial because the short- and long-term consequences of the pandemic on chronic diseases are still unknown.

### Funding

This work was supported by the Prince Sattam bin Abdulaziz University, project number (PSAU/2023/R/1445). The funders had no role in study design, data collection and analysis, decision to publish, or preparation of the manuscript.

### Grant Disclosures

The following grant information was disclosed by the authors:
The Prince Sattam bin Abdulaziz University: PSAU/2023/R/1445.

### Competing Interests

The authors declare there are no competing interests.

### Author Contributions

- Zeshan Ali conceived and designed the experiments, performed the experiments, analyzed the data, prepared figures and/or tables, and approved the final draft.
- Lienda Bashier Eltayeb analyzed the data, authored or reviewed drafts of the article, and approved the final draft.
- Sndos Z.A. Fattiny analyzed the data, prepared figures and/or tables, authored or reviewed drafts of the article, and approved the final draft.
- Iftikhar Younis Mallhi performed the experiments, authored or reviewed drafts of the article, and approved the final draft.
- Farah Javed conceived and designed the experiments, performed the experiments, authored or reviewed drafts of the article, and approved the final draft.
- Ali Abdullah Alyousef analyzed the data, prepared figures and/or tables, authored or reviewed drafts of the article, and approved the final draft.
- Qandeel Ijaz performed the experiments, analyzed the data, prepared figures and/or tables, and approved the final draft.
- Shoaib Younas conceived and designed the experiments, performed the experiments, authored or reviewed drafts of the article, and approved the final draft.
- Ishrat Khan conceived and designed the experiments, prepared figures and/or tables, and approved the final draft.
- Zahra Batool analyzed the data, prepared figures and/or tables, and approved the final draft.
- Muhammad Ahmad analyzed the data, prepared figures and/or tables, and approved the final draft.
## Human Ethics

The following information was supplied relating to ethical approvals (i.e., approving body and any reference numbers):

The study was conducted in accordance with the Declaration of Helsinki, and approved by the Institutional Review Board of Barani Institute of Sciences Sahiwal, Pakistan (IRB# 2001).

## Data Availability

The raw data are available in the Supplemental Files.

## Supplemental Information

Supplemental information for this article can be found online at http://dx.doi.org/10.7717/peerj.16730#supplemental-information.

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
