# Peer review of "COVID-19 and diet: efforts towards a balanced diet and sustainable nutrition among university students in Pakistan"

_PeerJ, doi:10.7717/peerj.16730_

## Round 0.1 · original submission · Minor Revisions

Please, authors kindly consider the comments raised by the reviewers to revise your work.

Kindly elaborate in detail in the manuscript, especially your responses to reviewers' comments.

**Language Note:** PeerJ staff have identified that the English language needs to be improved. When you prepare your next revision, please either (i) have a colleague who is proficient in English and familiar with the subject matter review your manuscript, or (ii) contact a professional editing service to review your manuscript. PeerJ can provide language editing services - you can contact us at copyediting@peerj.com for pricing (be sure to provide your manuscript number and title). – PeerJ Staff

·

Basic reporting

The manuscript presents interesting and important results. A few issues, however, need to be addressed:
Abstract
- Please provide the study period, methods of the study, and keywords.
Introduction
- Provide the situation update of COVID-19 in the study area.
- Line 76-78, “The current study … (Figure 1).”, the current study focus on knowledge, COVID-19, diet, and physical activity that different from the abstract (Line 21-23) address the awareness and diet. Please explain more about the current study.
- Figure 1, please provide the source of information.
Materials and methods
Participants
- Provide the sampling methods and sample size.
Results
Medical history of participants
- The authors should not repeat information similar to presented table 2.
- Table 2, please provide the total number of all participants.
Discussion
- Divide them into several paragraphs and each paragraph should contain sub-headings, which can enhance readability.
- Provide the suggestions for future research.
- Provide a little more contextual information about national food safety emergency response plans.

Experimental design

No comment.

Validity of the findings

No comment.

Additional comments

No comment.

Reviewer 2 ·

Basic reporting

Relevance of the study:
This investigation is original, generating some scientific interest, given the role of diet to promote health and considering that covid-19 significantly altered many aspects of human daily fife, including access to food.
Title:
The title is misleading because it seems too generic, therefore I think it is important to mention that this a localized study. In this way, I recommend to modify the title
COVID-19 and diet: efforts towards balance diet for sustainable nutrition among Pakistanese university students
Or
COVID-19 and diet: efforts towards balance diet for sustainable nutrition among university students in Pakistan

Abstract:
The abstract is well organized with a brief explanation of the purpose of the study followed by a summary of the experimental methodology followed. Then the authors presented the most relevant conclusions observed, and the abstract ends with a concluding remark, highlighting the most relevant conclusion, as a “message to take home”.
Nevertheless, I think that the abstract is too generic and does not provide any concrete results. So the authors should add some quantitative results.
For example: Results showed that university students of medical discipline were slightly more aware of immunity enhancing foods and nutritive values of foods as compared to the engineering students. – you should add the values of percentages or scores of these two types, for example – x% against y% respectively.
Do this for the major findings.
Introduction:
The introduction helps to frame and contextualize the work. It presents some state of the art on a number of topics which are essential to the work that was carried out, and it serves as a justification for its purpose, also highlighted at the end of introduction.
Materials and methods:
The description of the methodologies applied to obtain and treat the data are presented very clearly. I do not have any recommendations.

Results:
The results are well presented, for example in Tables and Figures which are very elucidative and allow a good understanding of the data obtained.


Discussion and Limitations:
The discussion is very comprehensive, with relevant references cited along the text. The limitations part is also relevant to frame the results of the research.

Conclusions:
The conclusions part is also well formulated, presenting the most relevant findings of the work.

Experimental design

Valid and I have no comments

Validity of the findings

I confirm validity of the findings though the use of adequate techniques to treat the data

Additional comments

No additional comments

---

## Round 0.2 · accepted · Accept

Having thoroughly examined the revised manuscript, I am very convinced that it is now acceptable for publication. Thank you authors for finding PeerJ your journal of choice, and look forward to your future scholarly contributions.

·

Basic reporting

I would like to express my appreciation for the revisions made by the authors in response to the comments. After careful review, I am pleased to recommend the acceptance of the manuscript.

Experimental design

None.

Validity of the findings

None.

Additional comments

None.